# N-O Ligand Supported Stannylenes: Preparation, Crystal, and Molecular Structures

**Hannah S. I. Sullivan** [1,2], **Andrew J. Straiton** [1], **Gabriele Kociok-Köhn** [3] and **Andrew L. Johnson** [1,*]

1    Department of Chemistry, University of Bath, Bath BA2 7AY, UK
2    Centre for Sustainable Chemical Technologies, University of Bath, Bath BA2 7AY, UK
3    Material and Chemical Characterisation Facility (MC²), University of Bath, Bath BA2 7AY, UK
*    Correspondence: a.l.johnson@bath.ac.uk

**Abstract:** A new series of tin(II) complexes (**1**, **2**, **4**, and **5**) were successfully synthesized by employing hydroxy functionalized pyridine ligands, specifically 2-hydroxypyridine (hpH), 8-hydroxyquinoline (hqH), and 10-hydroxybenzo[h]quinoline (hbqH) as stabilizing ligands. Complexes $[Sn(\mu-\kappa^2ON-OC_5H_4N)(N\{SiMe_3\}_2)]_2$ (**1**) and $[Sn_4(\mu-\kappa^2ON-OC_5H_4N)_6(\kappa^1O-OC_5H_4N)_2]$ (**2**) are the first structurally characterized examples of tin(II) oxypyridinato complexes exhibiting $\{Sn_2(OCN)_2\}$ heterocyclic cores. As part of our study, $^1$H DOSY NMR experiments were undertaken using an external calibration curve (ECC) approach, with temperature-independent normalized diffusion coefficients, to determine the nature of oligomerisation of **2** in solution. An experimentally determined diffusion coefficient (298 K) of $6.87 \times 10^{-10}$ m² s⁻¹ corresponds to a hydrodynamic radius of Ca. 4.95 Å. This is consistent with the observation of an averaged hydrodynamic radii and equilibria between dimeric $[Sn\{hp\}_2]_2$ and tetrameric $[Sn\{hp\}_2]_4$ species at 298 K. Testing this hypothesis, $^1$H DOSY NMR experiments were undertaken at regular intervals between 298 K–348 K and show a clear change in the calculated hydrodynamic radii form 4.95 Å (298 K) to 4.35 Å (348 K) consistent with a tetramer $\rightleftarrows$ dimer equilibria which lies towards the dimeric species at higher temperatures. Using these data, thermodynamic parameters for the equilibrium ($\Delta H° = 70.4\ (\pm 9.22)$ kJ mol⁻¹, $\Delta S° = 259\ (\pm 29.5)$ J K⁻¹ mol⁻¹ and $\Delta G°_{298} = -6.97\ (\pm 12.7)$ kJ mol⁻¹) were calculated. In the course of our studies, the Sn(II) oxo cluster, $[Sn_6(m^3-O)_6(OR)_4:\{Sn^{(II)}(OR)_2\}_2]$ (**3**) (R = $C_5H_4N$) was serendipitously isolated, and its molecular structure was determined by single-crystal X-ray diffraction analysis. However, attempts to characterise the complex by multinuclear NMR spectroscopy were thwarted by solubility issues, and attempts to synthesise **3** on a larger scale were unsuccessful. In contrast to the oligomeric structures observed for **1** and **2**, single-crystal X-ray diffraction studies unambiguously establish the monomeric 4-coordinate solid-state structures of $[Sn(\kappa^2ON-OC_9H_6N)_2)]$ (**4**) and $[Sn(\kappa^2ON-OC_{13}H_8N)_2)]$ (**5**).

**Keywords:** stannylene; 2-hydroxy-pyridine; 8-hydroxyquinoline; 10-hydroxybenzo[h]quinoline; molecular structures; DOSY NMR

## 1. Introduction

The chemistry of heavy carbene analogues [MR₂:M = Si, Ge, Sn, Pb; R = stabilizing anionic ligand) has attracted considerable interest for some time now and is of fundamental interest in main group p-block chemistry [1–3]. Their marked dual Lewis acidic and Lewis basic characteristics, alongside steric and electronic tuneability via astute ligand selection result in diverse and often unique chemistry [2].

A particular class of tetrylenes of interest are the stannylene complexes with the general formula $[Sn(R-L)_{2-x}(X)_x]$, where [R-L] is a monoanionic ligand bearing a lariat donor group (L) fragment capable of forming either intra- or inter-molecular Lewis base adducts with the metal atom, Figure 1, thus acting as an anionic chelate or bridging ligand, and where X is an ancillary monodentate ligand such as a halogen atom or $\{N(SiMe_3)_2\}$ group [4–9]. These Sn(II) compounds have potential applications as ligands for transition

metals, intermediates in organic synthesis [10–13], catalysts in the conversion of small molecules [14–20] or the ring opening polymerisation of lactones [21], or as precursors for electronic thin film materials [8,18,22–26].

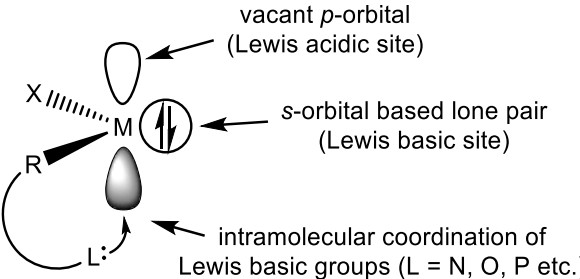

**Figure 1.** Stabilization of stannylene [Sn(R-L)$_{2-x}$(X)$_x$)] complexes.

It is well established that the nature of the ligand (donor atoms) has a significant effect on the chemical and structural properties of these heavy carbene analogues. Amongst the library of [Sn(II)(R-L)$_{2-x}$(X)$_x$] systems, perhaps the most frequently studied are those bearing an N-N' chelate as the [R-L] ligands, such as amidinates [22,27–29], guanidinates [8,25,30–33], β-diketiminates [5,21,34–39], and amino/imino-pyrrolides [9]. While Sn(II) [Sn(R-L)$_{2-x}$(X)$_x$] systems bearing an *N,O* chelate ligand such as amino alcohols [7,40–48], amino-phenols [49], organic amides [23,50], salicylaldiminato [50–53], ketiminate [54], and hydroxyquinolinates [55–64] are known, their derivative chemistry is less developed.

While the anionic derivatives of 2-hydroxypyridine (hpH) [65,66] and 8-hydroxyquinoline (hqH) [67,68] have been utilized as 1,3- 1,4-k$^2$*N,O* chelate ligands, respectively, for a range of metals, examples of the coordination chemistry of 10-hydroxybenzo[h]quinoline (hbqH) (Figure 2) and its derivatives are scarce, despite its commercial availability.

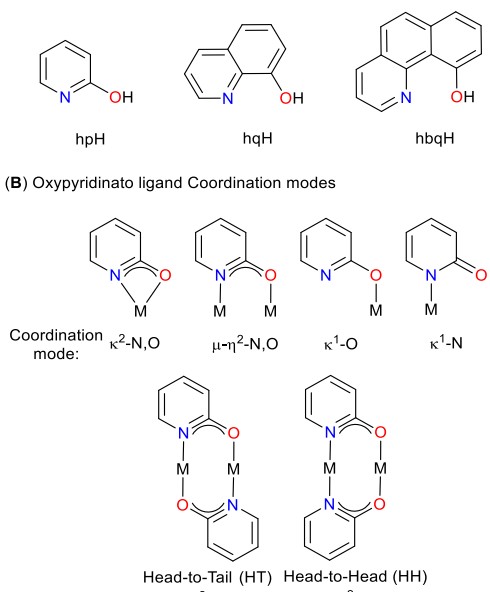

**Figure 2.** Overview of the *N,O* donor pro-ligands, hpH, hqH, and hbqH used in this study (**A**), and the common coordination modes observed for the 2-hydroxpyridinato (hp) ligand (**B**).

The anions of 2-hydroxypyridine derivatives belong to a wider class of tri-atomic bridging ligands, such as carboxylate, triazenide, amidinate, sulfonates, and phosphates [65,69]. The presence of two different donor atoms results in rich coordination chemistry with a variety of potential binding modes for the oxypyridinato ligand, common examples of which are shown in Figure 2, illustrating the unusual binding modes of the {hp} ligand

that constrained 1,3-*N,O* (four-membered ring) coordination can induce. A search of the literature reveals numerous oxypyridinato ligated transition metal complexes, whilst there are in comparison only a handful of main group complexes (reported in the CSCD) [70–75], and to date, none of these represent Sn(II)-supported species. In contrast, the quinolinato, {hq} and the oxybenzo[h]quinolinato {hbq} ligands exclusively display 1,4- and 1,5-$\kappa^2$-*N,O* coordination modes, respectively.

To-date, Sn(II) derivatives of these systems (i.e., {hp}, {hq} and {hbq}), are limited to a handful of homoleptic [61,64] and heteroleptic [55–60,62,63] Sn(II) hydroxyquinolate complexes. Here, we describe the details of the synthesis and structure of a series of Sn(II) complexes supported by the 1,3-, 1,4- and 1,5-$\kappa^2 N,O$ chelate ligands oxypyridinato (hpH), hydroxyquinolinato {hqH} and oxybenzo[h]quinolinato {hbq}, respectively.

## 2. Synthesis of Tin(II) Complexes

In all cases, the isolated products were characterised by solution-state NMR ($^1$H, $^{13}$C{$^1$H} and $^{119}$Sn{$^1$H}) spectroscopy and elemental analysis. The reaction of the bulky Sn(II) amide system [Sn{N(SiMe$_3$)$_2$}$_2$] with ligand 2-hydroxypyridine in a stoichiometric 1:1 reaction results in the formation and isolation after recrystallisation (−28 °C in toluene), of complex **1** as pale yellow crystals (Scheme 1). The $^1$H NMR spectrum of **1** (in C$_6$D$_6$) clearly shows the presence of a singlet resonance at δ = 0.37 ppm, representative of the presence of an {HMDS} group. The $^1$H NMR spectra also clearly show a series of four multiplets corresponding to the aromatic protons of the {hp} ligand at δ = 6.20, 6.48, 6.97 and 7.99 ppm, respectively, in a 1:1:1:1:18 ratio, with the {N(SiMe$_3$)$_2$} group indicating the formation of a mono-substituted complex of the general form [{hp}Sn{N(SiMe$_3$)$_2$}]. This is supported by the $^{13}$C{$^1$H} NMR spectra, which clearly show the expected six resonances [δ = 170.0, 142.6, 142.2, 116.6 114.0, and 5.9 ppm].

**Scheme 1.** Formation of complexes **1** and **2**.

Likewise, the $^{119}$Sn{$^1$H} NMR spectra shows a single sharp resonance [δ = –84.9 ppm], respectively, (*cf.* δ = 776.0 ppm, for [Sn{N(SiMe$_3$)$_2$}$_2$]). A reaction of [Sn{N(SiMe$_3$)$_2$}$_2$] with two equivalents of 2-hydroxypyridine in toluene results in the formation of a colourless precipitate, which, after heating into solution followed by filtration and cooling, afforded a crop of colourless crystals of **2**. The $^1$H NMR spectrum of **2** (in C$_6$D$_6$) shows only four multiplets corresponding to the aromatic protons of the {hp} ligand at δ = 6.02, 6.62, 6.92, and 7.63 ppm, respectively, in a 1:1:1:1 ratio. The clear absence of a peak corresponding to the {N(SiMe$_3$)$_2$} group indicates the formation of a *bis*-substituted complex of the general form [Sn{hp}$_2$]. The $^{13}$C NMR spectra consist of five resonances [δ = 112.7, 115.7, 140.4, 143.7, and 169.3 ppm], while the $^{119}$Sn NMR spectra display a single broad resonance at δ = −525.5 ppm. Low-temperature NMR studies (223 K) on complex **2** in d$_8$-Tol showed no significant changes in the $^1$H, $^{13}$C{$^1$H}, or $^{119}$Sn{$^1$H} NMR spectra, suggestive of an exchange process faster than the NMR time scale even at 223 K. Below 223 K, low solubility issues resulted in the loss of the $^{119}$Sn signal.

To determine the nature of the oligomerisation of **2** in solution, $^1$H DOSY NMR experiments were undertaken, using the novel external calibration curve (ECC) approach,

with temperature-independent normalized diffusion coefficients, for MW-determination described by Stalke et al. [76,77].

The room temperature (298 K) $^1$H DOSY NMR spectrum for compound **2**, shown in Figure 3, clearly shows a single diffusion coefficient ($D_{norm}$ = 6.87 × 10$^{-10}$ m$^2$ s$^{-1}$) which corresponds to a hydrodynamic radius of *Ca.* 4.95 Å.

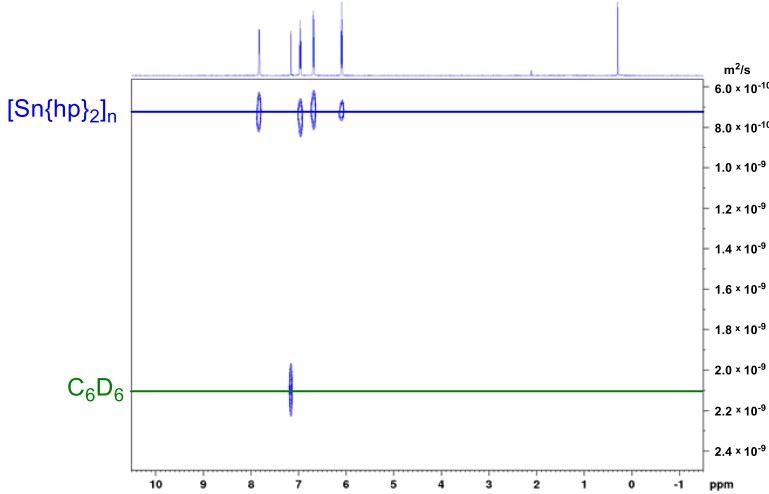

**Figure 3.** Plot showing the $^1$H DOSY NMR spectrum for compound **2** (298 K).

Based on the single-crystal data (*infra vide*), the molecular volume of the tetrameric **2** observed in the solid state was calculated as ca. 721 Å$^3$, with a molecular radius of 5.56 Å. While molecular shapes are important to the accurate interpretation of diffusion data, diffusion coefficient/molecular weight correlation studies have shown that merged calibration curves based around molecules in solution behave as either (i) compact spheres, (ii) dissipated spheres or ellipsoids, or (iii) expanded discs that can be used to estimate molecular weights.

Assuming each "[Sn{hp}$_2$]" unit is "spherical" and possesses an approximate volume of 183 Å$^3$, as determined by X-ray crystallography, the experimentally determined hydrodynamic volume, based on the Stokes-Einstein equation, is consistent with the presence of a trimeric species in solution. While a trimeric species is possible, we believe a more likely scenario is that the experimentally determined hydrodynamic volume is an averaged diffusion coefficient (and hydrodynamic volume) constant with a rapid (on the NMR timescale) equilibria between dimeric [Sn{hp}$_2$]$_2$ and tetrameric [Sn{hp}$_2$]$_4$ species (Scheme 2) at 296 K.

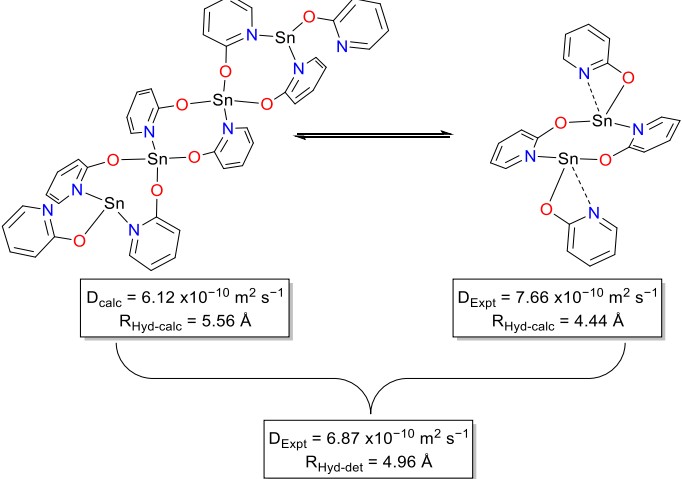

**Scheme 2.** Proposed Tetramer Dimer equilibria observed for **2**.

Using Stalke's method, and correcting empirically for molecular density (using an empirically determined density correction factor of 1.5535 and hence accounting for the heavy Sn atom) [78], we predict a molecular mass of 937 Da (*c.f.* a calculated mass for a trimeric species [Sn{hp}$_2$]$_3$ of 921 Da, 2% error). To test our hypothesis, $^1$H DOSY NMR experiments were undertaken at regular intervals between 298 K–348 K. Table 1 and Figure 4 show the diffusion coefficients of **2**, 5.43 mM in d8-Tol as a function of temperature. At 298 K, $^1$H DOSY NMR affords an experimentally determined diffusion coefficient of D = 7.97 × 10$^{-10}$ m$^2$ s$^{-1}$, which on warming to 348 K increases to D = 1.75 × 10$^{-9}$ m$^2$ s$^{-1}$. The observed change in the calculated hydrodynamic radii is concomitant with a decrease in the calculated hydrodynamic radii, such that at 348 K R$_{Hyd-cal}$ = 4.37 Å (*cf.* R$_{Hyd-cal}$ = 4.93 Å at 298 K and a crystal graphically determined R = 4.44 Å) consistent with a tetramer $\rightleftarrows$ dimer equilibria which lies towards the dimeric species at higher temperatures. When cooled to room temperature, the observed diffusion coefficient returns to D = 7.97 × 10$^{-10}$ m$^2$ s$^{-1}$, implying a low barrier to tetramer formation.

**Table 1.** Experimentally determined ($^1$H DOSY NMR) diffusion coefficients (D), calculated Hydrodynamic radii (R$_{Hyd-cal}$) and calculated Molecular volume (V$_{cal}$) of **2** at temperatures between 298–348 K.

| Temp (K) | D (m$^2$s$^{-1}$) | R$_{Hyd-cal}$ (Å) | V$_{cal}$ (Å$^3$) |
|---|---|---|---|
| 298 | 7.974 × 10$^{-10}$ | 4.93 | 503 |
| 308 | 9.416 × 10$^{-10}$ | 4.85 | 477 |
| 318 | 1.116 × 10$^{-9}$ | 4.70 | 436 |
| 328 | 1.310 × 10$^{-9}$ | 4.57 | 400 |
| 338 | 1.538 × 10$^{-9}$ | 4.41 | 360 |
| 348 | 1.750 × 10$^{-9}$ | 4.37 | 350 |

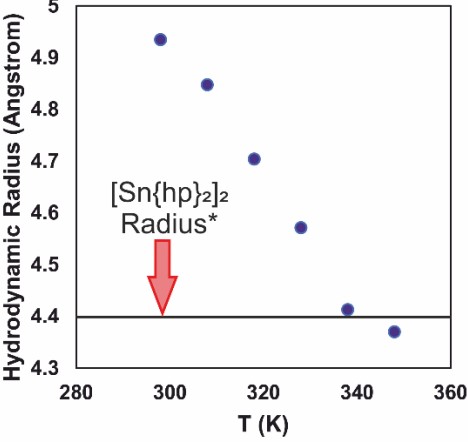

**Figure 4.** Graph showing the experimentally determined diffusion coefficients of **2**, 5.43 mM in d$^8$-Tol, as a function of temperature. *: Radius (4.44 Å) as determined by single-crystal X-ray diffraction.

A subsequent attempt followed to obtain thermodynamic parameters for the equilibrium process from the experimentally observed diffusion coefficients and the crystal graphically determined radii of the tetramer and the dimer species, which in turn can be used to estimate the molar fractions of the tetramer and dimer in solution and the equilibrium constants K at various temperatures (see ESI). A plot of ln(K$_{eq}$) vs. 1/T reveals an endothermic reaction process with the following parameters: ΔH° = 70.4 (±9.22) kJ mol$^{-1}$, ΔS° = 259 (±29.5) J K$^{-1}$ mol$^{-1}$, and ΔG°$_{298}$ = −6.97 (±12.7) kJ mol$^{-1}$.

While these values are only estimates, and no substantial emphasis should be placed on them, the values may be compared with those found for another dimer—tetramer equilibrium process [79] and indicate that in this instance, tetramer formation is significantly favoured at lower temperatures.

During our investigation into the synthesis of complex **2**, crystals of a second product, **3**, were serendipitously isolated from the reaction liquors. Unfortunately, the low yield of the product inhibited full characterisation. Attempts to produce crystals of **3** on a larger scale, either by a controlled reaction of [Sn{N(SiMe$_3$)$_2$}$_2$] with 2-hydroxypyridine and water, as shown in Scheme 3, or by the slow (4 weeks) diffusion of air and atmospheric moisture into a solution of **2** in toluene at room temperature, failed [80].

**Scheme 3.** Formation of the Sn(II) oxo cluster (**3**).

The single-crystal X-ray analysis of these crystals revealed the product to be an Sn(II) oxo cluster, specifically [{Sn$_6$($\mu_3$-O)$_6$($\mu$-OC$_5$H$_5$N)$_2$(OC$_5$H$_4$N)$_2$}:2{Sn($\kappa^2$ON-OC$_5$H$_4$N)$_2$}] (**3**), *vide infra*. Scheme 3 shows a possible synthetic pathway to **3**.

The reaction of [Sn{N(SiMe$_3$)$_2$}$_2$] with equimolar amounts of the ligands 8-hydroxyquinoline and 10-hydroxybenzo[h]quinoline, respectively (Scheme 4), results in the formation of the *bis*-quinolinato complexes, **4**–**5**, in yields < 50%, suggestive of an equilibrium in which the putative mono-amide intermediates are unstable with respect to disproportionation, and formation of the *bis*-complex.

**Scheme 4.** The synthesis of complexes **4** and **5**.

In both cases (**4** and **5**), the $^1$H and $^{13}$C{$^1$H} NMR spectra display a series of multiplet resonances in the aromatic region. The intrinsic $C_2$-symmetry of complexes **4**–**5** (Scheme 4) is negated somewhat in solution by a rapid, so-called "flip-flop" equilibrium process in which the N $\rightarrow$ Sn coordination bonds repeatedly open and close, a process observed for other Sn(II) systems containing *O,N* chelate ligands, resulting in a single set of resonances. The absence of a peak corresponding to the {N(SiMe$_3$)$_2$} group is consistent with the formation of the *bis*-substituted complexes [Sn{hq}$_2$] and [Sn{hbq}$_2$], an assertion supported by the presence in the $^{119}$Sn{$^1$H} NMR spectra of single peaks at $\delta$ = −386 ppm and −542 ppm, respectively. The stoichiometric reaction (2:1) of the pro-ligands hqH and hbqH, with [Sn{N(SiMe$_3$)$_2$}$_2$] produces the expected complexes cleanly in moderate to high yields (69–84%). The formation of the *bis*-quinolinato complexes was supported by elemental analysis.

### 3. Single-Crystal X-ray Diffraction Studies and Molecular Structures of Tin(II) Complexes

X-ray diffraction studies on single crystals of complexes **1**, **2**, **4**, and **5** unambiguously established their solid-state structures. The structures of the heteroleptic and homoleptic complexes, $[Sn(\mu\text{-}\kappa^2ON\text{-}OC_5H_4N)(N\{SiMe_3\}_2)]_2$ (**1**), and $[Sn_4(\mu\text{-}\kappa^2ON\text{-}OC_5H_4N)_6(\kappa^1O\text{-}OC_5H_4N)_2]$ (**2**) are shown in Figures 5 and 6, respectively, with selected bond lengths and angles in Tables 2 and 3, respectively. Compound **1** (Figure 5) exists as a centrosymmetric dimer in the solid-state, comprising two independent $[\{hp\}Sn\{N(SiMe_3)_2\}]$ molecules in the asymmetric unit cell, with symmetry operators generating the second half of each dimer.

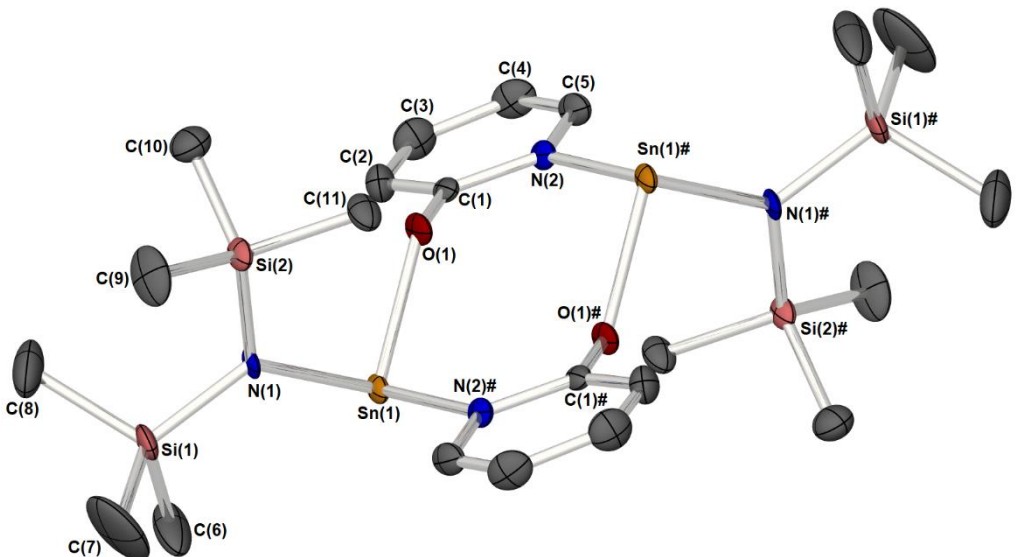

**Figure 5.** The molecular structure of one of the two full molecules of complex **1** present in the unit cell. Symmetry transformations (#) used to generate equivalent atoms; #: −X, −Y, 1−Z. Thermal ellipsoids are shown at 50% probability. Hydrogen atoms have been omitted for clarity.

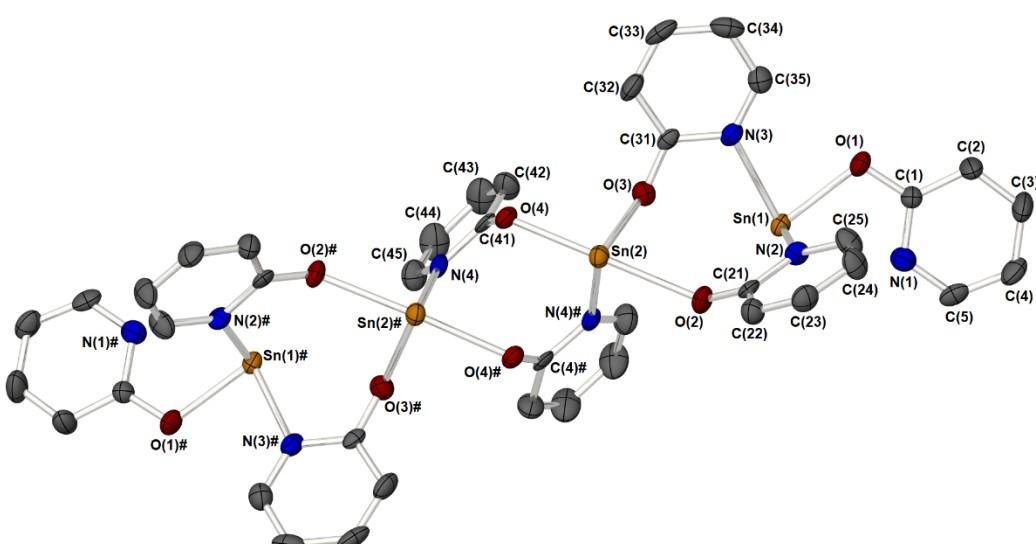

**Figure 6.** The solid-state molecular structure of complex **2**. Symmetry transformations (#) used to generate equivalent atoms; #: 3/2−X, Y, 1−Z. Thermal ellipsoids are shown at 50% probability. Hydrogen atoms have been omitted for clarity.

**Table 2.** Selected bond lengths (Å) and bond angles (°) for complex **1**. Symmetry-generated atoms are marked with (#).

| Selected Bond Lengths (Å) | | Selected Bond Angles (°) | |
|---|---|---|---|
| Sn(1)-N(1) | 2.1009(16) | N(1)-Sn(1)-N(2)# | 90.30(7) |
| Sn(2)-N(3) | 2.1064(16) | N(1)-Sn(1)-O(1) | 93.36(6) |
| | | O(1)-Sn(1)-N(2)# | 90.72(6) |
| Sn(1)-O(1) | 2.1491(14) | | |
| Sn(2)-O(2) | 2.1434(15) | N(3)-Sn(2)-N(4)# | 90.08(7) |
| | | N(3)-Sn(2)-O(2) | 92.91(6) |
| Sn(1)-N(2) | 2.2751(19) | O(2)-Sn(2)-N(4)# | 92.46(6) |
| Sn(2)-N(4) | 2.2590(19) | | |
| C(1)-O(1) | 1.302(3) | O(1)-C(1)-N(2) | 116.7(2) |
| C(1)-N(2) | 1.355(3) | O(2)-C(21)-N(4) | 116.6(2) |
| C(21)-O(2) | 1.302(3) | | |
| C(21)-N(4) | 1.351(3) | O(1)-C(1)-N(2)-Sn(1)# | 3.2(2) |
| | | O(2)-C(21)-N(4)-Sn(2)# | 3.0(2) |

**Table 3.** Selected bond lengths (Å) and bond angles (°) for complex **2**. Symmetry-generated atoms are marked with (#).

| Selected Bond Lengths (Å) | | Selected Bond Angles (°) | |
|---|---|---|---|
| Sn(1)-O(1) | 2.135(3) | N(2)-Sn(1)-N(3) | 86.64(13) |
| Sn(1)-N(2) | 2.239(4) | O(1)-Sn(1)-N(2) | 87.04(14) |
| Sn(1)-N(3) | 2.337(4) | O(1)-Sn(1)-N(3) | 79.58(13) |
| Sn(2)-O(2) | 2.370(3) | O(2)-Sn(2)-O(4) | 157.89(13) |
| Sn(2)-O(3) | 2.142(3) | O(3)-Sn(2)-N(4)# | 83.40(14) |
| Sn(2)-O(4) | 2.263(3) | O(2)-Sn(2)-O(3) | 78.57(12) |
| Sn(2)-N(4)# | 2.230(4) | O(2)-Sn(2)-N(4)# | 78.58(13) |
| | | O(3)-Sn(2)-O(4) | 86.85(12) |
| O(1)-C(1) | 1.322(5) | O(4)-Sn(2)-N(4)# | 83.35(12) |
| C(1)-N(1) | 1.333(6) | O(1)-C(1)-N(1) | 116.5(4) |
| O(2)-C(21) | 1.289(6) | O(2)-C(21)-N(2) | 116.1(4) |
| C(21)-N(2) | 1.357(6) | O(3)-C(31)-N(3) | 116.6(4) |
| O(3)-C(31) | 1.310(6) | O(4)-C(41)-N(4) | 116.4(4) |
| C(31)-N(3) | 1.347(6) | Sn(1)-O(1)-C(1)-N(1) | 6.8(5) |
| O(4)-C(41) | 1.297(6) | Sn(1)-N(2)-C(21)-O(2) | 0.6(5) |
| C(41)-N(4) | 1.348(6) | Sn(1)-N(3)-C(31)-O(3) | 2.3(5) |
| | | Sn(2)#-N(4)-C(41)-O(4) | 0.4(4) |

Both the oxygen and nitrogen donor atoms of the oxypyridinato ligands bridge the two *transoidal* Sn{N(SiMe$_3$)$_2$} moieties in a head-to-tail (HT) $\mu$-$\eta^2$ coordination mode, resulting in an eight-membered Sn$_2$O$_2$C$_2$N$_2$ cyclic core to the molecule.

Each Sn(II) centre in **1** is three-coordinate and may be regarded as being at the centre of a distorted trigonal pyramid in which the Sn atoms are coordinated by one oxygen atom and one nitrogen atom from two different {hp} ligands in a "Head-to-Tail" fashion (see Figure 2), as well as the nitrogen of a terminal {N(SiMe$_3$)$_2$} ligand [Sn(1)-N(1) 2.101(2) Å and Sn(2)-N(3) 2.106(2) Å]. The bond angles about each Sn(II) atom in **1** [i.e., N(2A)-Sn(1)-O(1) 90.71°; N(2A)-Sn(1)-N(1) 90.30°; O(1)-Sn(1)-N(1) 93.36°] suggest the absence of significant *sp*-hybridisation at tin.

The Sn–O and Sn–N bond lengths within the eight-membered ring [Sn(1)-O(2) 2.1491(14), Sn(1)-N(2) 2.2751(19), and Sn(2)-O(2) 2.1434(15), Sn(2)-N(4) 2.2590(19) Å] are shorter than the range expected for simple Sn ← O and Sn ← N Lewis acid–base complexes [9,51,81–83]. Although the pyridine rings are stacked in the crystal lattice in an antiparallel-displaced

arrangement, the inter-ring distance (>4 Å) is greater than that expected for any significant electronic interaction (i.e., 3.4 Å) [84].

Furthermore, in each Sn-containing eight-membered ring, the core of the molecule adopts a chair-like ring conformation similar to that observed in the pyridine-selenolate complex [Sn$_2$\{$\mu$-$\eta^2$Se-2-C$_5$H$_4$N\}$_2$\{SeC$_5$H$_4$N\}$_2$] [85]. While Sn(II) oxypyridinato complexes such as **1** have not, to the best of our knowledge, been previously reported, the core structure of **1** is reminiscent of Sn(II) carboxylate complexes [\{Sn(NR$_2$)($\mu$-$\eta^2$-O$_2$CAr)$_2$\}] (Ar = *m*-terphenyl) [86]. As the bridging ligand oxypyridinato is isolobal to a carboxylate ligand [O$_2$CR], this comparison is unsurprising. Similarly, **1** is also reminiscent of the *bis*(trimethylsilyl)amido Sn(II) triflate [\{Sn(NR$_2$)($\mu$-$\eta^2$-OTf)$_2$\}] complex, which also consists of an Sn-containing eight-membered heterocyclic ring [83]. However, unlike the triflate derivative, complex **1** is a discrete entity and shows no evidence of intermolecular Sn···O interactions between adjacent eight-membered rings. The puckering of the central \{Sn$_2$O$_2$C$_2$N$_2$\} ring may be considered to be a result of the folding of the eight-membered ring along the two inter-ligand O···N vectors, such that it adopts a chair-like conformation (**I**) (Scheme 5). The extent of the folding ($\theta_{ring}$) may be defined as the angle between the \{SnON\} planes and the \{O$_2$N$_2$\} plane. In the absence of steric interactions, the "ideal" folding angle ($\theta_{ring}$) should be ca. 130°; the fold angle in **1** is $\theta_{ring}$ = 129.14°. A similar bonding arrangement for the $\mu_2$-O,N hydroxypyridine ligand is observed in the aluminium complex [($^t$Bu)$_2$Al($\mu$-O-2-C$_5$H$_4$N)]$_2$ [70].

**Scheme 5.** Schematic drawing showing the coordination of the \{hp\} ligand in complex **1**.

The molecular structure of **2** is shown in Figure 6 and comprises a series of three eight-membered *spiro*-Sn$_2$O$_2$C$_2$N$_2$ rings that link through the Sn centres to create a one-dimensional three-link chain; selected bond lengths and angles are given in Table 3. Compound **2** is tetrameric, containing four Sn atoms and eight oxypyridinato ligands with three different coordination modes; two \{hp\} ligands are observed in terminal positions coordinating to the two three-coordinate Sn(II) centres via the oxygen atom of the \{hp\} ligand. Two oxypyridinato ligands in the central eight-membered ring bridge two four-coordinate Sn(II) centres in a $\mu$–$\eta^2$ head-to-tail (HT) coordination mode. The four remaining oxypyridinato ligands bridge between the two four-coordinate Sn(II) centre in the central 8-membered ring and the two three-coordinate Sn(II) centres in a $\mu$–$\eta^2$ head-to-head (HH) coordination mode.

At the centre of complex **2** is a \{Sn$_2$O$_2$C$_2$N$_2$\} ring comparable in coordination geometry to that observed in complex **1**; the central ring in complex **2** can be considered to be in a boat-like conformation (**II**) (Scheme 6), resulting from folding along Sn···O vectors such that the two pyridine rings tend towards co-planarity, with an angle between the \{SnONC$_5$\} planes of 35.36° and an inter-ring distance of 3.760(2) Å. As a result of this puckering, the Sn···Sn distance in the inner ring is marginally reduced to 4.082 Å (*cf.* 4.468 Å in **1**).

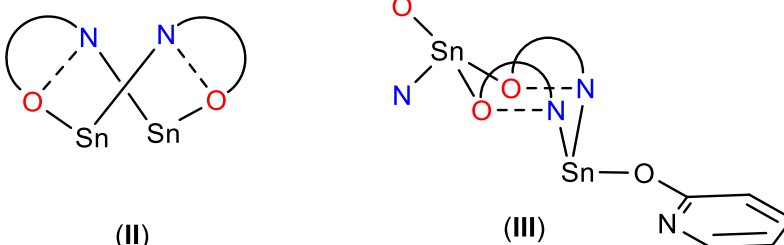

**Scheme 6.** Schematic drawing showing the coordination of the {hp} ligands in complex **2**.

Both Sn(II) centres in this central ring are four-coordinate, and may be considered to be the spiro atoms linking the three eight-membered {SnN$_2$C$_2$O$_2$Sn} rings together. While the two Sn(II) centres are four-coordinate, their geometry may be regarded as either distorted trigonal bipyramidal with a stereo-active lone pair of electrons or disphenoidal.

As noted above, the two outer {SnN$_2$C$_2$O$_2$Sn} rings are different to the central ring with the oxypyridinato ligands bridging the two Sn(II) centres in the outer rings in head-to-head bridging modes. The puckering of the "outer" Sn$_2$O$_2$C$_2$N$_2$ rings may be considered to be chair-like (**III**) (Scheme 6), resulting from the folding of the eight-membered ring along the two inter-ligand O···N vectors; θ$_{ring}$ angle between {SnO$_2$} plane/{O$_2$N$_2$} plane = 39.91(5)°, and between {O$_2$N$_2$} plane/{SnN$_2$} plane = 97.92(5)°, thus resulting in a solid-state arrangement, such that the three-coordinate Sn(II) centres are folded under the {O$_2$N$_2$} plane (see (**III**) in Scheme 6).

The Sn–O and Sn–N bond lengths within both the three eight-membered rings and the exo-Sn-O bonds are within expected ranges for these complexes. Similarly, bond lengths and bond angles within the oxypyridinato ligands are all similar and within parameters observed in related complexes (*vide supra*).

While for complex **1** there are no comparable mono thio- or seleno-pyidinato complexes with which the coordination chemistry of oxy-pryidinato {hp} ligand can be compared, the homoleptic *bis*-thio and *bis*-seleno- complexes, [Sn{S-C$_5$NH$_4$}$_2$] [87] and [Sn{Se-C$_5$NH$_4$}$_2$] [85] are known, and display different structural chemistry to their oxygen-containing cousin, **2**. Similarly, the homoleptic lead(II) 3,5-dinitro-2-pyridonate complex possesses a polymeric 1D solid-state structure, consisting of a {Pb$_2$O$_2$} core [73].

As noted above, during our investigation into the synthesis of complex **2**, crystals of a second product were isolated from the reaction liquors. The single-crystal X-ray analysis of these crystals revealed the product to be the Sn(II) oxo cluster, [{Sn$_6$(μ$_3$-O)$_6$(μ-OC$_5$H$_5$N)$_2$(OC$_5$H$_4$N)$_2$}:2{Sn(κ$^2$ON-OC$_5$H$_4$N)$_2$}] (**3**), which is composed of a central {Sn$_6$O$_6$} core supported by four {hp} ligands, two of which are coordinated in *N,O*-μ-η$^2$ coordination mode, and two of which are terminal (Scheme 6). The cluster is further appended by the coordination of two [Sn{hp}$_2$] moieties, previously unobserved in this study, via a dative O←Sn interaction between two {μ$_3$-O} groups of the {Sn$_6$O$_6$} cluster and the Sn atoms of [Sn{hp}$_2$] units (Scheme 3).

While the principal {Sn$_6$O$_6$} core is comparable to other Sn(II) oxo-alkoxide systems [88], to the best of our knowledge, this is the first example of any such a cluster engaging in exo-coordinating to any metal centre via a dative M ← {μ$_3$-O} interaction, rather than an Sn(II) centre [89–92]. The molecular structure of complex **3** is shown in the supplementary information, along with selected bond lengths and bond angles.

The solid-state molecular structures of **4** and **5**, as determined by single-crystal X-ray crystallography, are shown in Figure 7, with selected bond distances (Å) and bond angles (°) in Table 4. Complex **3** crystallises in orthorhombic space group *Pbcn* with half of the molecular complex in the asymmetric unit. Contrastingly, complex **5** crystallises in the triclinic space group *P-1*, with one full molecule in the unit cell alongside one whole molecule of disordered toluene.

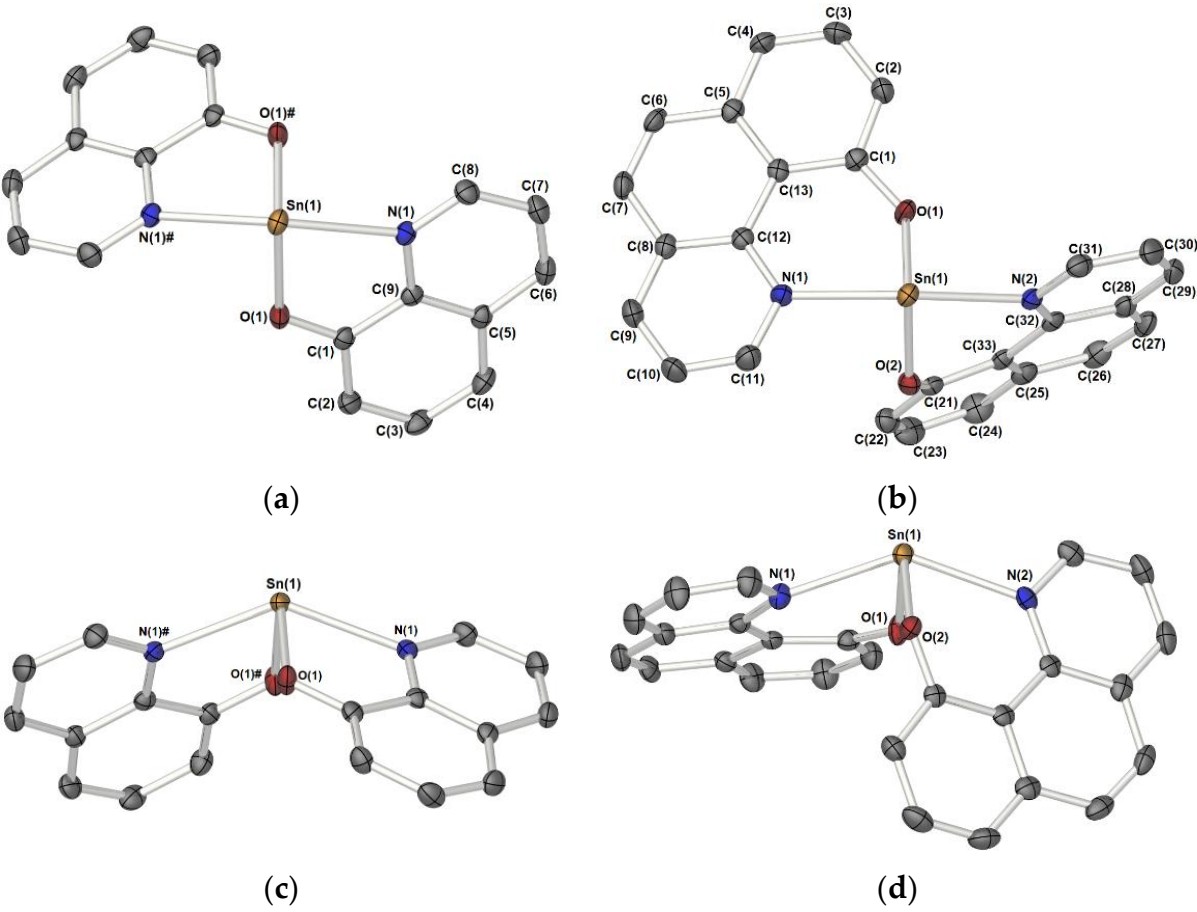

**Figure 7.** The molecular structure of **4** (**a**,**c**) and **5** (**b**,**d**). Thermal ellipsoids are shown at 50% probability. H-atoms are omitted for clarity. Solvent of crystallization (toluene) in **5** is omitted for clarity. Symmetry transformations used to generate equivalent atoms: #1 3/2−X, Y, 1−Z.

**Table 4.** Selected bond lengths (Å) and bond angles (°) for complexes **4** and **5**.

| 4 | | 5 | |
|---|---|---|---|
| Selected Bond Lengths (Å) | | | |
| Sn(1)-O(1) | 2.0985(17) | Sn(1)-O(1) | 2.0824(15) |
| | | Sn(1)-O(2) | 2.0779(17) |
| Sn(1)-N(1) | 2.4014(19) | Sn(1)-N(1) | 2.4140(19) |
| | | Sn(1)-N(2) | 2.4289(19) |
| O(1)-C(1) | 1.326(3) | O(1)-C(1) | 1.328(3) |
| | | O(2)-C(21) | 1.326(3) |
| Selected bond angles (°) [1] | | | |
| N(1)-Sn(1)-N(1)# | 138.54(9) | N(1)-Sn(1)-N(2) | 141.52(7) |
| O(1)-Sn(1)-O(1)# | 98.23(10) | O(1)-Sn(1)-O(2) | 97.33(7) |

[1] # denotes symmetry generated atom (#: 3/2−X, Y, 1−Z).

The solid-state structures of **4** and **5** are consistent with the spectroscopic data, suggesting the monomeric structures of the complexes in the solid-state are retained in solution. Both complexes have what may be described as "two-bladed propeller"-like geometry, with a $C_2$ axis lying on the bisectors of the N–Sn–N axis. Whilst **3** and **4** are both chiral, by virtue of their molecular $C_2$ symmetry, the enantiomers co-crystallise in non-centrosymmetric space groups.

Despite coordination numbers of four, in both **4** and **5**, analysis of the N-Sn-N and O-Sn-O bond angles about the Sn(II) centres suggests the Sn atoms adopt a pseudo trigonal

bipyramidal, or disphenoidal geometry, each with a stereoactive lone pair of electrons [**4**: $\tau$ = 0.67, **5**: $\tau$ = 0.74] with the nitrogen atoms occupying the axial coordination sites and oxygen atoms the equatorial positions. The N–Sn–N bond angles increase from ~138.5° to 141.5° as the chelate ring is changed from 1,4-$\kappa^2 N,O$, to 1,5-$\kappa^2 N,O$ coordination, respectively. The associated O–Sn–O angles [**4**: 98.224(17)°, **5**: 97.331(14)°] are both close to 90°, suggesting that the Sn–O bonds are largely based around the Sn(II) *p*-orbitals. The 1,4-$\kappa^2 N,O$ and 1,5-$\kappa^2 N,O$ coordination of the ligands are associated with acute bite angles (N–Sn–O) of 73.23(2)° for **4** and 74.76(3)/77.07(3)° for **5** and induce a distortion of the axial vector (N–Sn–N) from linearity to angles of 138.56(2)° and 141.52(5)° for **4** and **5**, respectively.

In complexes **4** and **5**, the Sn–O distances (**4**: 2.0985(17) Å; **5**: 2.0824(15) and 2.0779(17) Å) are slightly shorter than comparable Sn–O interactions in complexes **1** and **2**. For **4**, the Sn-O and Sn-N bond lengths are comparable to related systems (*vide supra*). Contrastingly, the Sn–N interactions in **4** and **5** (2.4014(19) Å and 2.4140(19)/2.4289(19) Å) are slightly longer than those observed in **1** and **2**. However, both Sn-O and Sn-N bond lengths are in good agreement with the interatomic distances reported for comparable Sn(II) O-N chelate complexes (*vide supra*).

As noted previously, complex **4** possesses molecular $C_2$ symmetry. In contrast, one of the {hbq} ligands is folded along the inter-ligand O(1)···N(2) vectors, such that the three-rings of the {hbq} ligand are moved away from the second {hbq} ligand. A closer inspection of the intermolecular distances associated with complex **5** reveals both favourable $\pi$-$\pi$ stacking interactions, between the {C(1)-N(1)} and {C(1)#-N(1)#} poly-aromatic rings, [3.687(12)Å between the parallel planes defined by the atoms {C(1)-N(1)} and {C(1)#-N(1)#}, respectively], and C-H···$\pi$ interactions between adjacent molecules as shown in Figure 8, [H(4)-(1)$_{cent}$ = 2.658(12) Å and H(6)-(2)$_{cent}$ = 3.227(12) Å], which are presumably the origin of the observed distortion from molecular $C_2$ symmetry in the solid state. Interestingly, complex **5** appears to be one of a small number of structurally characterised examples of the {hbq} ligand coordinated to a metal centre, other than [Be{hbq}$_2$] [93] [Me$_2$Ga{hbq}] and [Me$_2$In{hbq}]$_2$ [94], as reported in the literature.

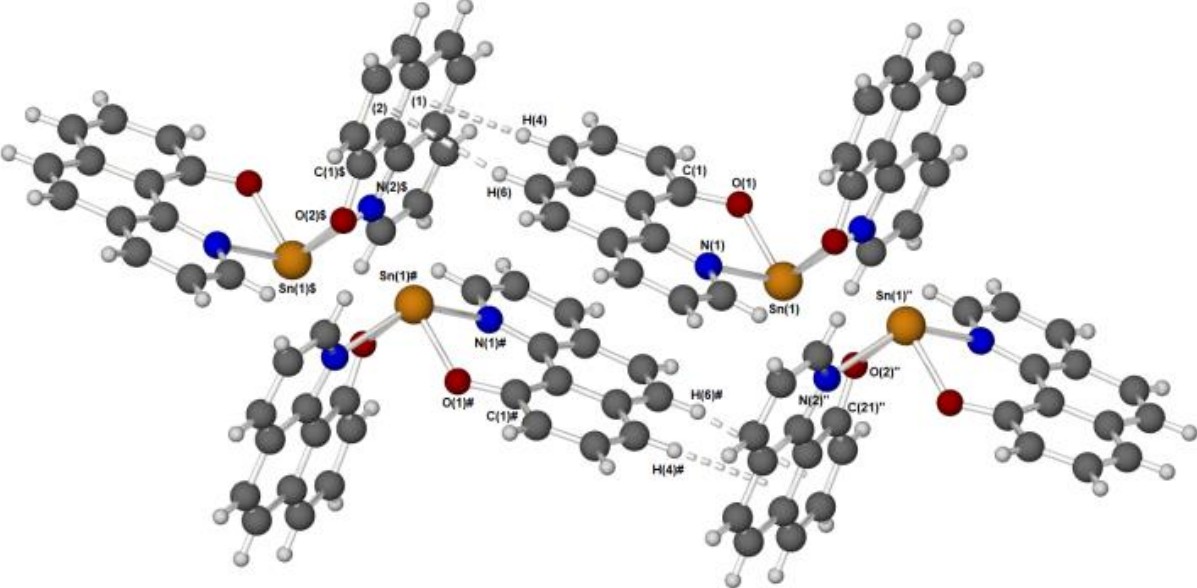

**Figure 8.** A representation of the H-Bonding and $\pi$-$\pi$ stacking observed in the solid-state structure of **5**. Bond lengths: H(4)-(1)$_{cent}$ = 2.658(12) Å and H(6)-(2)$_{cent}$ = 3.227(12) Å. Inter-plane distance: {C(1)-N(1)}/{C(1)#-CN(1)#} = 3.687(12) Å. Symmetry transformations used to generate equivalent atoms: #1: X−1, 1+Y, -Z; $#: X−1, 1+Y, Z;"# X−1, 1+Y, Z.

## 4. Experimental Section

**Experimental Details**: [1]H, [13]C, and [119]Sn NMR spectra were recorded on Bruker Avance 300 or 500 MHz FT-NMR spectrometers, as appropriate, in saturated solutions at room temperature. Chemical shifts are expressed in ppm with respect to $Me_4Si$ ([1]H and [13]C) and SnMe4 ([119]Sn) in $C_6D_6$. Elemental Analysis was performed externally by London Metropolitan University Elemental Analysis Service.

All reactions were carried out under an inert atmosphere using standard Schlenk techniques. The solvents were dried and degassed under an argon atmosphere over activated alumina columns using an Innovative Technology solvent purification system (SPS). The Sn(II) amide, [Sn{N(SiMe_3)_2}_2], was prepared by the literature methods [95].

**Synthesis of [Sn(µ-κ²ON-OC_5H_4N)(N{SiMe_3}_2)]_2 (1):** To a cooled solution of [Sn{N(SiMe_3)_2}_2] (1 mmol, 0.44 g) in toluene (10 mL) a suspension of 2-hydroxypyridine (1 mmol, 0.10 g) in toluene (10 mL) was added and stirred for 2 hrs. The resulting solution gradually became yellow in colour. The reaction mixture was heated to near reflux and filtered through a Celite® pad (hot) to afford a clear pale-yellow filtrate. The removal of excess solvent under vacuum followed by storage at −28 °C yielded 0.20 g pale yellow microcrystals crystals (Yield = 54%). Elemental Analysis $(C_{11}H_{22}N_2OSi_2Sn)_2$ (expected): C 35.59% (35.40%), H 6.02% (5.94%), N 7.51% (7.51%). 1H NMR ($C_6D_6$, 500 MHz): δ 0.37 (s, 18H, -N{SiMe_3}_2), 6.19–6.23 (m, 1H, 5-CH), 6.47–6.50 (m, 1H, 3-CH), 6.94–6.99 (m, 1H, 4-CH), 7.97–8.01 (m, 1H, 6-CH). [13]C NMR ($C_6D_6$, 125.7 MHz): δ 6.0 (-N{SiMe_3}_2), 114.0 (5-C), 116.6 (3-C), 142.2 (4-C), 142.6 (6-C), 170.0 (2-C). [119]Sn NMR ($C_6D_6$, 186.5 MHz): δ −85.

**Synthesis of [Sn_4(µ-κ²ON-OC_5H_4N)_6(κ¹O-OC_5H_4N)_2] (2):** To a cooled solution of [Sn{N(SiMe_3)_2}_2] (2 mmol, 0.88 g) in toluene (10 mL) a suspension of 2-Hydroxypyridine (4 mmol, 1.23 g) in toluene (10 mL) was added. A colourless precipitate was observed to form over time. The suspension was allowed to stir for 2 h before the precipitate was heated into solution. Filtration through a Celite® pad afforded a colourless filtrate. Slow cooling (−28 °C) afforded 0.58 g of small starburst colourless crystals. (yield = 94%). Elemental Analysis $(C_{10}H_8N_2O_2Sn)_4$ (expected): C 38. 7% (39.14%), H 2.85% (2.63%), N 9.11% (9.13%). [1]H NMR ($C_6D_6$, 500 MHz): δ 5.98–6.05 (m, 1H, 5-CH), 6.54–6.61 (m, 1H, 3-CH), 6.88–6.95 (m, 1H, 4-CH), 7.58–7.67 (m, 1H, 6-CH). [13]C NMR ($C_6D_6$, 125.7 MHz): δ 112.71 (5-C), 115.70 (3-C), 140.44 (4-C), 143.73 (6-C), 19.30 (2-C). [119]Sn NMR ($C_6D_6$, 186.5 MHz): δ −526.

**Synthesis of [Sn(µ-κ²ON-OC_9H_6N)_2] (4):** A suspension of 8-Hydroxyquinoline (4 mmol, 0.5806 g) in toluene (10 mL) was added, via cannula, to a cooled solution of [Sn{N(SiMe_3)_2}_2] (2 mmol, 0.88 g) in toluene. A bright yellow precipitate was generated instantly. The suspension was allowed to stir for 2 h before the precipitate was heated into solution and filtered through a Celite® pad affording a yellow filtrate. Slow cooling (−28 °C) afforded 0.62 g of large yellow crystals. (yield = 76%). Elemental Analysis $C_{18}H_{12}N_2O_2Sn$ (expected): C 53.27% (53.12%), H 3.02% (2.97%), N 6.85% (6.88%). [1]H NMR ($C_6D_6$, 500 MHz): δ 8.32 (d, $J^3_{H\text{-}H}$ = 5 Hz, 1H 2-CH) 7.45 (d, $J^3_{H\text{-}H}$ = 10 Hz, 1H, 7-CH) 7.19 (dd, $J^3_{H\text{-}H}$ = 10 Hz, $J^3_{H\text{-}H}$ = 10 Hz, 1H, 3-CH), 7.10 (dd, $J^3_{H\text{-}H}$ = 10 Hz, $J^3_{H\text{-}H}$ = 5 Hz, 1H, 6-CH), 6.69 (d, $J^3_{H\text{-}H}$ = 10 Hz, $J^3_{H\text{-}H}$ = 10 Hz, 1H, 4-CH), 6.61 (dd, (dd, $J^3_{H\text{-}H}$ = 5 Hz, $J^3_{H\text{-}H}$ = 5 Hz, 1H, 5-CH). [13]C NMR ($C_6D_6$, 125.7 MHz): δ 112.6 (4-C), 115.2 (6-C), 121.0 (5-C), 130.6 (3-C), 130.8 (8-C), 138.4 (7-C), 140.2 (9-C), 143.3 (2-C), 162.9 (1-C). [119]Sn NMR ($C_6D_6$, 186.5 MHz): δ −386.

**Synthesis of [Sn(µ-κ²ON-OC_13H_8N)_2] (5):** A suspension of 10-Hydroxybenzo[h]quinoline (4 mmol, 0.7810 g) in toluene (10 mL) was added, via cannula, to a cooled solution of [Sn{N(SiMe_3)_2}_2] (2 mmol, 0.88 g) in toluene. A bright yellow precipitate was generated instantly. The suspension was stirred for 2 h before the precipitate was heated into solution and filtered through a Celite® pad affording a yellow filtrate. Slow cooling (−28 °C) afforded 0.62 g of large yellow crystals. (yield = 85%). Elemental Analysis $C_{26}H_{16}N_2O_2Sn$:$C_7H_8$ (expected): C 65.97% (66.04%), H 4.11% (4.03%), N 4.65% (4.66%). [1]H NMR ($C_6D_6$, 500 MHz): δ 6.67 (dd, $J^3_{H\text{-}H}$ = 10 Hz, 5 Hz, 1H, CH), 6.83 (dd, $J^3_{H\text{-}H}$ = 5 Hz, 2 Hz, 2H, CH), 6.91 (dd, $J^3_{H\text{-}H}$ = 10 Hz, 10 Hz, 1H, CH), 6.99–7.03 (m 2H CH), 7.22 (d, $J^3_{H\text{-}H}$ = 2 Hz, 1H, CH), 8.08 (dd, $J^3_{H\text{-}H}$ = 5 Hz, 2 Hz, 1H, CH), 8.79 (dd, $J^3_{H\text{-}H}$ = 5 Hz, 2 Hz,

1H, CH). $^{13}$C NMR ($C_6D_6$, 125.7 MHz): δ 116.9, 118.3, 120.0, 120.3, 120.7, 123.0, 125.6, 126.3, 126.7, 130.6, 145.0 145.6, 169.3. $^{119}$Sn NMR ($C_6D_6$, 186.5 MHz): δ −542.

**Single Crystal X-ray Diffraction Studies:** Experimental details relating to the single-crystal X-ray crystallographic data for compounds **1**–**4** are summarized in Table S1 (ESI). The single-crystal X-ray crystallography data were collected at 150 K on RIGAKU Super-Nova Dual wavelength diffractometer equipped with an Oxford Cryostream, featuring a micro source with MoKα radiation (λ = 0.71073 Å) and Cu Kα radiation (λ = 1.5418 Å). Crystals were isolated from an argon-filled Schlenk flask and immersed under oil before being mounted onto the diffractometer. The structures were solved by direct methods throughout and refined on F2 data using the OLEX2 suite of programs [30]. All hydrogen atoms were included in idealized positions and refined using the riding model. Refinements were straightforward with no additional points that merit note. CCDC 2015978–2015982 contains the supplementary crystallographic data for this paper. These data can be obtained free of charge at www.ccdc.cam.ac.uk/conts/retrieving.html (accessed on 25 July 2022) [or from the Cambridge Crystallographic Data Centre, 12, Union Road, Cambridge CB2 1EZ, UK; Fax: + 44-1223/336-033; E-mail: deposit@ccdc.cam.ac.uk].

## 5. Conclusions

In recent years, 1,3-O,N, 1,4-O,N, and to a lesser extent 1,5-O,N chelate ligands based around the "basic" oxo-pyridinato system have been shown to play an important role as sterically and electronically tuneable ligands to a range of metal centres. The ambidentate nature of these tuneable ligands has allowed the stabilisation of metals spanning most of the periodic table. However, there is a paucity of main group metal examples bearing such ligands. Here, we describe an explorative investigation into the chemistry of these systems and highlight the array of reactive and novel bonding modes such versatile ligands display in a family of Sn(II) complexes bearing the pyridinato ligands oxopyridinato {hp} (**1** and **2**), quinolinato, {hq} (**3**) and oxybenzo[h]quinolinato {hbq} (**4**). The Sn(II) complexes bearing these ligands were synthesized and characterised in solution and in the solid state. In the case of the 1,3-$\kappa^2$NO oxpyridinato systems, the ligand displays a diverse coordination geometry, mirroring some observations noted in transition metal chemistry. In contrast, the coordination chemistry of the quinolinato, {hq}, and oxybenzo[h]quinolinato {hbq} ligands display a much more limited variety in their bonding modes, with **3** and **4** displaying distorted trigonal bipyramidal geometries. To the best of our knowledge, complexes **1** and **2** are the first reported examples of Sn(II) centres supported by oxo-pyridinato ligands. In addition, complex **5** is a rare example of a coordination complex of the oxybenzo[h]quinolinato (**5**) ligands, and as such, one of only a small number from across the whole of the periodic table.

**Supplementary Materials:** The following supporting information can be downloaded at: https://www.mdpi.com/article/10.3390/inorganics10090129/s1, Figure S1: Two views of the molecular structure of complex **3**; Table S1: Selected Bond lengths [Å] for Complex **3**; Table S2: Selected Bond angles [°] for complex **3**; Table S3: Crystal data and structure refinement for complexes **1**–**5**.

**Author Contributions:** Conceptualization, A.L.J.; methodology, H.S.I.S.; Crystallographic analysis, A.L.J. and G.K.-K.; NMR Investigation, A.J.S.; writing—original draft preparation, A.L.J.; writing—review and editing, A.L.J. and A.J.S.; supervision, A.L.J.; project administration, A.L.J.; funding acquisition, A.L.J. All authors have read and agreed to the published version of the manuscript.

**Funding:** This research was funded by EPSRC, grant number EP/L0163541 and EP/G03768X/1.

**Institutional Review Board Statement:** Not applicable.

**Informed Consent Statement:** Not applicable.

**Data Availability Statement:** All the data reported in the study can be found in the supporting information and from the CCDC repository with accession numbers: 2015978–2015982.

**Acknowledgments:** We thank the EPSRC Doctoral Training Centre in Sustainable Chemical Technologies (CSCT) for a PhD studentship (H.S.I.S).

**Conflicts of Interest:** The authors declare no conflict of interest.

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
