# Peer review of "N-O Ligand Supported Stannylenes: Preparation, Crystal, and Molecular Structures"

_inorganics, doi:10.3390/inorganics10090129_

Round 1

Reviewer 1 Report

I would suggest a MAJOR REVISION of this manuscript, including English language. See more details in attached file.

Author Response

We thank the reviewer for their comments and suggested amendments.  In all cases the suggested amendments and changes have been made. We hope that these are sufficient.

Reviewer 2 Report

The manuscript inorganics-1859537 entitled “N-O ligand supported Stannylenes: Preparation, Crystal, and 2 Molecular Structures” describes the synthesis of a series of four new tin(II) complexes bearing N,O chelating ligands and the structure of a Sn(II) oxo cluster containing oxypyridinato ligands. All complexes were characterised by NMR spectroscopy, including DOSY experiments, elemental analysis and single-crystal X-ray diffraction.

The manuscript is interesting, well written and the conclusions are supported by experimental data. The manuscript can be accepted for publication after addressing the following points:

11)      Scheme 1 (line 100) – please correct [Sn{N(SiMe3}2] to [Sn{N(SiMe3)2}2]

22)      Table 1 (line 180) – please use the same font in all the table

33)      Maybe it would be better to move the Sn(II) oxo cluster discussion here, as it is referred as compound 3. It can be confusing for the reader to go from compound 2 to 4, and then 3only  appears towards the end of the manuscript. Another option would be to introduce a sentence stating that during crystallization of 2 a second product was isolated that was only characterized by SCXRD and point to the discussion in that section (section 3).

44)      Please increase figures 4 and 5 resolution as they appear to be blured.

55)      Figure 6 (line 367) – please correct (a) and (b) – bottom – to (c) and (d). Also, “Solvent of crystallization (toluene) in 4” should be corrected to “...in 5”

66)      Line 399 – “suggests a distorted trigonal bipyramidal geometry about both centres [4: t = 0.67, 5: t = 0.74]” – these complexes are tetracoordinated and not pentacoordinated, therefore a trigonal bipyramidal geometry is not possible. Please explain.

Author Response

We thank the reviewer for their helpful comments.
in all case the comments have been addressed and changes made to the manuscript as suggested by the reviewer.
The reviewer did wish us to address the point made below:
“suggests a distorted trigonal bipyramidal geometry about both centres [4: t = 0.67, 5: t = 0.74]” – these complexes are tetra-coordinated and not penta-coordinated, therefore a trigonal bipyramidal geometry is not possible. Please explain.
we have added to the text of the manuscript to explain that despite 4-coordinate geometries in complexes 4 and 5, with the presence of a stereo-active lone pair of electrons the complexes adopt a pseudo TBP geometry. we hope that this amendment is sufficient.

Reviewer 3 Report

Comments to authors:

Manuscript ID: inorganics-1859537

Johnson and co-workers have reported “N-O ligand supported Stannylenes: Preparation, Crystal, and Molecular Structures”. The work presented here is interesting and impacts to appeal to a wide readership. The manuscript is overall well-structured and documented nicely with satisfactory data. I recommend this manuscript in its present format for publication in Inorganics.

Author Response

We thank the reviewer for their comments.

Round 2

Reviewer 1 Report

I would suggest a MINOR REVISION of this manuscript. See more details in attached file.

Author Response

The changes suggested by reviewer 1 have been made:
i) a full stop has been added the first sentence.
ii) the two sentences concerning compound 3 have been re constructed the remove any confusion.
iii) the formula has been removed from the caption for clarity.
iv) pmm has been changed to ppm
Kind regards
ALJ